# Quantitative Evaluation of Stem-like Markers of Human Glioblastoma Using Single-Cell RNA Sequencing Datasets

**DOI:** 10.3390/cancers15051557

**Published:** 2023-03-02

**Authors:** Yue He, Kristina B. V. Døssing, Ane Beth Sloth, Xuening He, Maria Rossing, Andreas Kjaer

**Affiliations:** 1Department of Clinical Physiology and Nuclear Medicine & Cluster for Molecular Imaging, Copenhagen University Hospital—Rigshospitalet & Department of Biomedical Sciences, University of Copenhagen, 2200 Copenhagen, Denmark; 2Computational and RNA Biology, University of Copenhagen, 2200 Copenhagen, Denmark; 3Center for Genomic Medicine, Rigshospitalet, Copenhagen University Hospital, 2100 Copenhagen, Denmark; 4Department of Clinical Medicine, University of Copenhagen, 2200 Copenhagen, Denmark

**Keywords:** glioblastoma stem cells, GBM stem-like markers, quantitative evaluation, single-cell RNA sequencing, CD133, SOX2, CD24, CD15

## Abstract

**Simple Summary:**

A common issue in glioblastoma stem cells (GSCs) studies is the need to efficiently and precisely target GSCs using reliable biomedical markers. Using single-cell RNA sequencing datasets, we quantitatively evaluated an extensive number of GSCs markers with multiple parameters that dictate the feasibility of various laboratory and therapeutic applications. We present promising marker candidates with their scores on the corresponding parameters and apply sequential selection based on these parameters. Both previously approved and novel markers are proposed according to the evaluation. We demonstrate the possibility of choosing a biomedical marker in a nonarbitrary way and provide quantitative references for potential GSCs markers.

**Abstract:**

Targeting glioblastoma (GBM) stem-like cells (GSCs) is a common interest in both the laboratory investigation and clinical treatment of GBM. Most of the currently applied GBM stem-like markers lack validation and comparison with common standards regarding their efficiency and feasibility in various targeting methods. Using single-cell RNA sequencing datasets from 37 GBM patients, we obtained a large pool of 2173 GBM stem-like marker candidates. To evaluate and select these candidates quantitatively, we characterized the efficiency of the candidate markers in targeting the GBM stem-like cells by their frequencies and significance of being the stem-like cluster markers. This was followed by further selection based on either their differential expression in GBM stem-like cells compared with normal brain cells or their relative expression level compared with other expressed genes. The cellular location of the translated protein was also considered. Different combinations of selection criteria highlight different markers for different application scenarios. By comparing the commonly used GSCs marker CD133 (*PROM1*) with markers selected by our method regarding their universality, significance, and abundance, we revealed the limitations of CD133 as a GBM stem-like marker. Overall, we propose *BCAN*, *PTPRZ1*, *SOX4*, etc. for laboratory-based assays with samples free of normal cells. For in vivo targeting applications that require high efficiency in targeting the stem-like subtype, the ability to distinguish GSCs from normal brain cells, and a high expression level, we recommend the intracellular marker *TUBB3* and the surface markers *PTPRS* and *GPR56*.

## 1. Introduction

Glioblastoma multiforme (GBM) is the most common aggressive brain cancer, with a poor prognosis, a median survival of 14 months, and only one in 20 patients being alive after five years [1,2]. Since the current Standard of Care (SoC) was introduced in 2005 [3] with macroradical surgery, external radiation therapy, and temozolomide therapy, there have been no major changes in the therapy or in the poor prognosis [4,5]. One of the major challenges to avoiding tumor recurrence is stem-like cells. These are cells in the GBM bulk tumor that possess the capacity for self-renewal and tumorigenesis [6,7]. After the surgical removal of the bulk tumor and treatment with chemo- or radiotherapy, any potentially stem-like cell residues left are likely to develop into a recurrent tumor [8,9,10]. Recent studies based on single-cell RNA sequencing (scRNA-seq) technology uncovered the GBM “stem-like” cells through gene set enrichment analysis featuring enrichment terms including “nervous system development” and “gliogenesis” [11], similar to the biological features of neural-progenitor cells. This resemblance is due to the fact that the GBM stem-like cells share many marker genes with somatic neural progenitor cells.

Before the era of scRNA-seq, much effort was devoted to discovering GBM stem-like cells and their biomedical markers in order to therapeutically target these cells. Several markers have become recognized over the years, such as CD133 (*PROM1*) [12,13], *SOX2* [14,15], CD24 [16], and CD15 [17]. Among these markers, CD133 has obtained the most attention so far. CD133 was initially identified as a protein bound to CD34 hematopoietic stem and progenitor cells [18]. The tumorigenic capability of CD133 positive cells was confirmed by both in vitro sphere formation [12,19,20] and in vivo xenograft assays [20,21]. However, some other studies claimed that there was a lack of robustness when using CD133 as a cancer stem cell marker [22,23].

Generally, there is a lack of direct comparison between the proposed stem-like markers for GBM, as most studies are independent and investigate only one or a few markers using different methods. scRNA-seq allows the identification of all the possible markers for the GBM stem-like subtype [11,24,25], avoiding the process of “trial and error”. However, the markers discovered by this approach are too many and indistinguishable to be applied in targeted assays that only allow a limited number of markers. In addition, in clinical settings, only a few markers can be applied at a time. Therefore, it is crucial to develop a pipeline to find the best markers among the many marker genes identified by scRNA-seq data.

The aim of this study is to quantitatively evaluate GBM stem-like markers identified by publicly available scRNA-seq data through multiple reality-relevant parameters: the universality and significance of GBM stem-like markers, the ability to distinguish GBM stem-like cells from normal brain cells, the expression level, and the cellular location of the translated protein. Different combinations of parameters are applied to reduce the number of candidates for different application requirements. With stringent standards, we propose the intracellular marker *TUBB3* and the cell-surface markers *PTPRS* and *GPR56* due to their excellent score for all parameters.

## 2. Materials and Methods

### 2.1. Data Acquisition

This study involves 37 GBM samples from three SMART-seq2 [26] based studies. All the data were obtained from the Gene Expression Omnibus (GEO) database. In total, there are 1091 cells from Darmanis et al. (GSE84465) [25], 7930 cells from Neftel et al. (GSE131928) [11], and 875 cells from Patel et al. (GSE57872) [24]. In addition, there are 982 normal brain cells including oligodendrocyte progenitor cells (OPCs), oligodendrocytes, vascular cells, neurons, astrocytes, and microglia from two studies (GSE67835, GSE84465) [25,27].

### 2.2. Preprocessing

The GBM cells from Darmanis et al. [25] and Patel et al. [24], and normal brain cells from Darmanis et al. [25,27], were processed from FASTQ files, going through FastQC quality control [28], Trimmomatic processing [29], and cell filtration with adaptive criteria that filter out cells with a low sum of reads and detected features (genes) and cells with excessive mitochondrial gene reads [30]. The cell filtration was conducted using the “isOutlier” function from Scater [30]. After filtration, 913, 557 GBM cells and 863 normal brain cells were left. These datasets were normalized using “library-size-normalization’ [30]. The data obtained from Neftel et al. [11] are in the form of a normalized count matrix in Transcript Per Million (TPM); all the steps before the cell filtration were already conducted by the author, therefore, only the cell filtration was applied to these data, with 7781 GBM cells passing the criteria. For all the datasets, genes that have total reads, summed up from all the cells, of less than 100 and mitochondria genes starting with “MT-” in their names were discarded. The preprocessed data were integrated with the corresponding cell metadata in the SingleCellExperiment object for further analysis [31].

### 2.3. Clustering and Enrichment Analysis

Clustering was conducted for each sample using the Leiden method from igraph [32], and the markers of each cluster were identified using Scanpy [33]. The marker genes for each cluster were first selected according to the criteria: logFC (log fold change) > 2 and *p*-value < 0.05, followed by extraction of the top 200 genes ranked by their *p*-value. The top marker genes were used for enrichment analysis using g:Profiler [34]. A cluster was considered “stem-like” if it possessed similar enrichment results to those of the “neural-progenitor-cell-like 1 (NPC-like 1)” or “neural-progenitor-cell-like 2 (NPC-like 2)” from the study by Neftel et al. [11].

### 2.4. Calculation of Abundance and Percentage-Rank

The abundance of a certain gene was defined as the percentage of cells expressing this gene in the same sample, indicating the abundance of cells that express the gene across the sample (Equation (Equation 1)). In order to correctly integrate data from different studies, we adopted a rank-based method to overcome batch effect. The percentage-rank of a gene was defined to indicate its expression level relative to other genes within a cell, calculated by the following steps: all the genes with non-zero reads in a cell were ranked in ascending order; the rank of the investigated gene was normalized by the number of genes with non-zero reads in the same cell and multiplied by 100% (Equation (Equation 2)). The exclusion of all the zero reads from the ranking was performed to avoid a biased ranking caused by cell number differences between samples, as samples with more cells would also cause the inclusion of more genes. The inclusion of more genes would result in a higher proportion of zero reads in each cell, in turn increasing the rank value for all non-zero reads.
(1)$$Abundance\;x_{a}=\frac{No.\;of\;cells\;that\;express\;gene\;x\times 100\%}{No.\;of\;cells\;from\;sample\;a}$$
(2)$$Percentage\;-rank\;x_{i}=\frac{rank(gene\;x\;in\;cell\;i)\times 100\%}{No.\;of\;non\text{-}zero\;genes\;in\;cell\;i}$$

### 2.5. Data Visualization

All the scatter plots, box plots, and bar plots were generated using ggplot2 [35]. The volcano plots were made using EnhancedVolcano [36]. The TSNE plots were generated using plotReducedDim from Scater [30]. The brain illustration was created using BioRender.com.

## 3. Results

### 3.1. GBM Stem-like Cluster Identification

Through clustering and enrichment analysis, we identified 28 stem-like clusters out of the total 92 clusters across the 37 GBM samples. The typical enrichment results of NPC-like 1 and NPC-like 2 are presented in Appendix A. The attributes of the NPC-like 1 and 2 subtypes indicated by the enrichment results are typical for neural stem cells. All of the enrichment analyses were based on the top 200 marker genes ranked by *p*-value after the preselection with the criteria: log fold change (logFC) > 2 and *p*-value < 0.05.

### 3.2. *PROM1* Is the Marker Gene for Eight out of 28 Total Stem-like Clusters with Moderate Significance

In the following sections, we will be using the gene name *PROM1* for CD133. *PROM1* serves as a marker gene for eight clusters out of the 28 total stem-like clusters, and the eight clusters belong to eight different samples (Figure 1). In summary, *PROM1* was found to be significantly overexpressed in 28.6% of the stem-like clusters. According to the volcano plot (Figure 1), *PROM1* ranks in the top 9.4%, 20.6%, 25.7%, 35.3%, 51.8%, 58.2%, 82.2%, and 93.2% (by *p*-value) among all the overexpressed markers (the selected zone on the right of the volcano plot Figure 1 by the criteria: logFC > 2 and *p*-value < 0.05) for the eight clusters, respectively. Overall, the significance of *PROM1* as a marker gene is not the most outstanding among all the markers of the eight clusters.

### 3.3. Proposing Multiple Standards for Choosing the Optimal GBM Stem-like Markers According to the Application

Through previous studies and our own investigations, it is believed that *PROM1* features the stem-like subtype in GBM [37,38]. However, the fact that only eight out of the total 28 stem-like clusters are marked by *PROM1* and its relatively low significance among all the markers led us to search in a wider range for other options that might target the GBM stem-like subtype better.

Our study featured 2173 unique stem-like marker candidates combined from the top 200 marker genes of the 28 stem-like clusters (Figure 2a). The principal filtration step is to select markers with high specificity to the stem-like subtype, indicated by the frequency of a gene being a marker gene (for the 28 stem-like clusters) and its significance (represented by median ranked *p*-value) (Figure 2b). The remaining markers could be further narrowed down with two optional approaches. One is to select markers that are significantly overexpressed in GBM stem-like cells compared with normal brain cells. The other is to select markers that exhibit higher expression levels relative to other genes expressed by the same cell (quantified by percentage-rank), and meanwhile are overexpressed compared with non-stem-like GBM clusters (quantified by logFC). The second approach finds markers suitable for assays that require a high expression level for their efficiency. Finally, the location of the expressed markers was also considered as it relates to the feasibility of certain applications (examples provided in the discussion section). The following section presents the markers selected by the principal selection step in combination with different optional criteria.

### 3.4. Selecting Frequent and Significant GBM Stem-like Markers

The frequency for each of the 2173 candidates to be a marker gene for a stem-like cluster was normalized by the total number of stem-like clusters and multiplied by 100%. The *p*-values of the marker genes were ranked ascendingly within each stem-like cluster, normalized by the number of marker genes for the cluster, and multiplied by 100%. The smaller the median *p*-value rank, the more significant the marker (indicated by the x-axis in Figure 2a. The median was taken across the 28 stem-like clusters). A higher value on the y-axis indicates a higher frequency of the gene being the marker gene for a stem-like cluster (Figure 2b). Based on these two parameters, the markers in the upper left corner of (Figure 2b) are optimal for specifying the stem-like subtype from other GBM cells. We applied the criteria: frequency > 14% and median *p*-value rank < 50% to obtain markers in this zone. This filtration step was passed by 251 marker genes. These 251 marker genes were used for further selection with the two optional approaches, as described previously.

In search for the most significant and frequently shared markers by the GBM stem-like clusters, *BCAN*, *SOX4*, *PTPRZ1*, *GPM6A SOX11*, *MAP2*, *TUBB2B*, *NREP*, *PTPRS*, *TUBB3*, *TUBA1A*, *DBN1*, *OLIG1*, *FXYD6*, *PMP2*, *SEMA5A*,*MLLT11*, *ASCL1*, *S100B*, and *MAGED1* are among the best candidates (Figure 2b). They are each found to be the marker for between 46% and 64% of the 28 stem-like clusters, and appear in a high significance range. Considering both parameters, they are superior to 99% of the total 2173 preselected stem-like markers. The markers in the lower right corner of Figure 2b are less representative of the GBM stem-like clusters. The most prominent markers selected by this method, such as *BCAN*, are expressed universally by the GBM stem-like cells. In comparison, *PROM1* is expressed sparsely by the same group of cells (Figure 2f).

### 3.5. Selecting Stem-like Markers Overexpressed by GSCs Relative to Normal Cells

The expression levels of the 251 selected marker genes were compared between GBM stem-like cells and normal brain cells. The median percentage-rank of each marker for all the GBM stem-like cells and for all the normal brain cells was used to calculate *p*-value and logFC, as shown in Figure 2c. The *p*-values were obtained by applying the Wilcoxon rank-sum test to the GBM stem-like cells and the normal cells group.

Among the 251 candidates, *C8orf46*, *FAM115A*, *GPR56*, *HMP19*, *LPPR1*, *MAGED4B*, *MLLT4*, *NGFRAP1*, *PTCHD2*, and *SEPT7* (red crosses in Figure 2d) were not expressed by the normal cells, indicating specificity to GBM stem-like cells compared with normal brain cells. Among these candidates, *GPR56* outperforms the others in the frequency–significance selection (Figure 2d). Because zero reads were excluded from rankings and did not enter the ranking-based comparison, they are not shown in Figure 2c.

Most of the remaining 241 markers present considerably low *p*-values in the comparison. The *p*-value criteria shown in (Figure 2c) is at 0.01, and all the highlighted markers still have*p* values far below it. *PTPRS* and *TUBB3* were found to remarkably distinguish GBM stem-like cells from normal brain cells while remaining favorable in the frequency–significance selection (Figure 2d; they are both orange genes and also appear in the upper left corner of the frequency–significance selection). The cancer-specificity of *TUBB3* is explicitly shown in Figure 3d, in comparison with the general progenitor cell marker *BCAN*. In fact, all the markers colored orange or brown in Figure 2c can be considered to distinguish GBM stem-like cells from normal brain cells. The advantageous markers from the cancer–normal comparison are also marked on the frequency–significance figure, to show the options that excel in both selections (Figure 2d).

### 3.6. Selection of GBM Stem-like Markers Based on Their Expression Level

The expression level of a marker can be of crucial importance for some targeting assays [39]. Therefore, we present the relative expression level represented by percentage-rank and logFC (from the differential expression analysis among GBM clusters), for the 251 genes that passed the frequency–significance selection (Figure 4). The median percentage-rank across all GBM stem-like cells and the median logFC over all the GBM stem-like clusters were plotted together. Markers that are expressed at a high level compared with other genes in the same cells and meanwhile overexpressed by the stem-like clusters within GBM are shown (Figure 4). The markers that can distinguish GBM stem-like cells from normal cells are also marked in the same figure (Figure 4) to show the combined results.

Among the 20 markers selected by the frequency–significance plot, *BCAN*, *PTPRZ1*, *PMP2*, *GPM6B*, *TUBB3*, and *S100B* outperform the others in the expression-level selection (Appendix A).

*TUBB3* excels in all three selections (frequency–significance selection (Figure 2b), the distinction to normal cells (Figure 2c), and expression level selection (Figure 4). This means that *TUBB3* is highly specific to the stem-like subtype within GBM and commonly found for the stem-like subtype, distinguishes cells from healthy brain cells, and is expressed at a sufficiently high level for ligand binding. Because all the markers used in this selection have a logFC greater than 2, which is considered significant in biological comparisons, the requirement shown by the y-axis of Figure 4 could have been lowered if more choices were needed.

### 3.7. The Location of a Marker Protein Should Be Considered to Achieve Successful Targeting

As well as the criteria involved in the selection steps described above, the cellular location of the protein translated from the corresponding marker gene is also crucial for the feasibility of using the marker in various applications [40]. Generally, it is easier to target markers located in the cell membrane than intracellular markers. Among the 251 selected marker candidates from the frequency–significance selection, about half of them are expressed on the cell membrane (marked as blue dots in Figure 2b, marker names given in Appendix A). Of all the markers that distinguish the GBM stem-like subtype from normal cells (orange, brown dots, and red crosses in Figure 2d), *PTPRS*, *ATP1A3*, *MAGED4*, *NNAT*, *ASIC4*, *ITGA7*, *GPR56*, *HMP19*, *LPPR1*, *MAGED4B*, and *PTCHD2* are cell membrane markers. If we apply more stringent criteria with all the previously mentioned parameters, then *PTPRS* and *GPR56* stand out as highly representative stem-like, cancer-specific, highly expressed surface markers. They only have a relatively lower logFC compared with *TUBB3* (Figure 4). Information regarding the location of the proteins encoded by the markers was identified in the Human Protein Atlas database [41] (refer to proteinatlas.org).

### 3.8. The Abundance and Expression Level of Selected GSCs Markers across the Samples

For some of the promising markers selected above, we examined their “abundance” (the proportion of cells that express non-zero values of the gene within a sample) and their percentage-rank across samples. The median abundance of PROM1 across samples is 18%. In comparison, *BCAN*, *PTPRZ1*, *SOX4*, and *GPM6A*, as representative markers from the frequency–significance selection, exhibit median abundances of 75%, 97%, 84%, and 88%, respectively. As a marker of prominent cancer-specificity, *LDHB* has a median abundance of 97%. The three recommended markers, *TUBB3*, *PTPRS*, and *GPR56*, by all standards exhibit a median abundance of 68%, 94%, and 86% (Figure 5).

The percentage-rank of each selected marker was also shown across the samples. Among all the cells that express the marker genes: *BCAN*, *PTPRZ1*, *SOX4*, *GPM6A*, *LDHB*, *TUBB3*, *PTPRS*, and *GPR56*, the percentage-rank is over 50% for 92%, 96%, 74%, 96%, 86%, 90%, 75%, and 87% of the cells, respectively, all higher than the 60% for *PROM1* (Figure 5). The median percentage-rank within each sample is higher than 50% for 100%, 97%, 73%, 100%, 81%, 88%, 77%, and 96% for the markers mentioned above. For *PROM1*, this value is 59% (Figure 5).

## 4. Discussion

The core strategy of this study is the use of nonparametric values such as frequency, percentage-rank, and *p*-value-rank in order to be able to integrate data from different studies. It allows us to quantitatively evaluate different aspects of a marker based on various patients from different studies [42,43,44]. Gene-expression data from different studies cannot be directly combined to draw comparative or statistical conclusions without eliminating the batch effects between them. However, most batch-correction tools either presume the data distribution or subtyping and inevitably introduce biases to the original data [45]. Therefore, we adopted the straightforward and robust rank-based method to integrate datasets [46,47]. Furthermore, the percentage-rank defined in this study has an advantage from its definition compared with gene counts. In a more explicit manner, it represents the expression level of a gene in comparison with all the other genes of the cell. Conversely, a normalized gene count does not provide much information in itself without conducting differential expression analysis. We believe that the rank-based data are more reliable and more relevant for a biomarker evaluation study. Furthermore, we believe that the most optimal way to validate their robustness is by presenting their universality among patients from different scRNA-seq-based datasets, together with their statistical significance. This is because bulk RNA sequencing data (such as TCGA sequencing data), which presents an average expression of each gene from all the cells in the sample, cannot be used for GSC marker identification validations.

Our identification of the NPC-like 1 and 2 subtypes can be verified by comparing our enrichment result (Appendix A) with the enrichment provided by Neftel et al. [11]. Both NPC-like 1 and 2 subtypes exhibit typical neural progenitor cell features. As a validation for our results, many other studies identified the same marker genes for GSC as we highlighted in Figure 2b; see Table 1 for a list of methods and references.

It was revealed that some cells identified as stem-like by their transcriptome profiles do not express *PROM1* (Figure 5). This provides an explanation for the tumorigenesis ability of CD133 negative cells reported by multiple studies [22,63]. Bhaduri et al. also discovered the “sparse” expression of *PROM1* [54]. We concluded that being CD133 positive is a sufficient but not necessary condition for being a GBM stem-like cell.

In favor of our findings of the normal brain-cell-distinctive GBM stem-like markers, seven out of eight total orange markers were reported to be overexpressed in different cancers (Figure 2c). *RCN1* was shown to be overexpressed in GBM stem-like cells compared with normal tissue [64]. *METTL7B* and *MAGED4* were discovered to be overexpressed prognostic markers in various gliomas [65,66,67,68,69]. *LDHB*, *PTPRS*, *UHRF1*, and *TUBB3* were proposed as universal prognostic and malignancy markers across many cancer types [70,71,72,73,74,75,76,77,78,79]. Studies regarding the differential expression of ATP1A3 in cancer tissue have not been found. The overexpression of multiple “red cross” markers (Figure 2d) was also reported in various cancer types by previous studies [61,80,81]. It should be clarified that most currently applied GBM stem-like markers are not normal-cell-distinctive, as they are also expressed by neural progenitor cells. For instance, previous studies suggested that CD133 can be used to separate stem cells from not only cancerous but also normal tissue [22,23,82], including brain [83]. Likewise, SOX2, CD15, and CD24 have also been identified as neural stem-cell markers [84,85]. Indeed, the denotation “stemness” refers to a particular biological feature of preserved multi-potency and self-renewal ability [86], which is not relevant to malignancy. To address this concern, we recommend focusing on the orange-, brown-, and red-cross-marked GBM stem-like markers in Figure 3d, to distinguish normal brain cells from GBM stem-like cells in clinical targeting.

We quantified the “stemness” of the total 2173 preselected GBM stem-like markers based on their universality (frequency) and significance as a basic step, to maximize the efficiency of the markers in targeting GBM stem-like subtypes within tumors and among patients. This was followed by two alternative selections to further obtain the options that can either (1) exclude normal cells during targeting, or (2) markers that are expressed at a high level by the targeted subtype, or both. Furthermore, the location of proteins encoded by the selected markers was also considered. Some targeting technologies prefer markers found in the cell membrane to intracellular markers, due to the difficulty of crossing membranes of live cells with targeting agents, such as targeted protein drugs [40]. In the end, we confirmed that the representative markers selected by our methods exhibit relatively higher abundance and expression levels across samples compared with *PROM1*.

*TUBB3* outperforms the other candidates with the combination of the first three selections but is expressed intracellularly. *PTPRS* and *GPR56* are cell-surface markers that stand out in the first two selections with slightly lower, but sufficient, expression levels. The stemness of these three markers is supported by the Neftel et al. study that identified them as marker genes for NPC1 or NPC2 subtypes [11]. Their cancer-specificity is supported by studies that reported their overexpression in various cancer types [61,71,72,73,77,78,79]. Although we started with a high number of candidates, only a few markers excelled at all the selections. Therefore, it is recommended to use only relevant parameters after the basic frequency–significance selection. For instance, efficient in vivo radionuclide-conjugated antibody targeting requires high expression of the biomarker in the targeted subtype [39], and the capability to exclude normal cells if the targeting agent is distributed ubiquitously in the brain. Surface markers are not compulsory in this case [39]. The optimal marker for this application is *TUBB3*, which is also found to be a marker for high-grade gliomas [87]. For in vitro isolation of glioblastoma stem cells from a bulk tumor, representative (determined by frequency–significance selection), and highly expressed surface markers are preferred, in which case cancer-specificity can be compromised. *PTPRZ1* is the most suitable marker in this scenario. Immunohistochemistry (IHC) assays require sufficiently expressed, highly representative stem-like markers, while the other two criteria are not as crucial. Preferable markers for IHC assays include *BCAN*, *PTPRZ1*, *PMP2*, *GPM6B*, *TUBB3*, and *S100B*. The choice of markers should be customized with relevant parameters according to the specific needs of the study in question. More application scenarios with suggested GSC markers are summarized in Appendix A.

## 5. Conclusions

Targeting glioblastoma multiforme (GBM) stem-like cells (GSCs) is the major motive of this study. Using parameters that quantify the universality, significance, expression level, and cancer-specificity of the candidate markers, we successfully compared 2,173 candidate GBM stem-like markers using single-cell RNA sequencing data. Our analyses suggest 20 markers, *BCAN*, *SOX4*, *PTPRZ1*, *GPM6A*, *SOX11*, *MAP2*, *TUBB2B*, *NREP*, *PTPRS*, *TUBB3*, *TUBA1A*, *DBN1*, *OLIG1*, *FXYD6*, *PMP2*, *SEMA5A*, *MLLT11*, *ASCL1*, *S100B*, and *MAGED1*, as the most universal and significant GSC markers across patients. Among these 20 stem-like markers, *BCAN*, *PTPRZ1*, *PMP2*, *GPM6B*, *TUBB3*, and *S100B* are expressed at high levels.

Comparing GSCs with normal brain cells, we found the markers *LDHB*, *RCN1*, *PTPRS*, *METTL7B*, *UHRF1*, *MAGED4*, *ATP1A3*, and *TUBB3* to have outstanding cancer-specificity. Thus, they are recommended for developing GSCs targeting agents for patient applications.

In conclusion, taking all the parameters and markers into account, *TUBB3*, *PTPRS*, and *GPR56* outperform the other candidates. We propose them as markers for the future targeting of GSCs in a wide variety of clinical applications. 

## Figures and Tables

**Figure 1 cancers-15-01557-f001:**
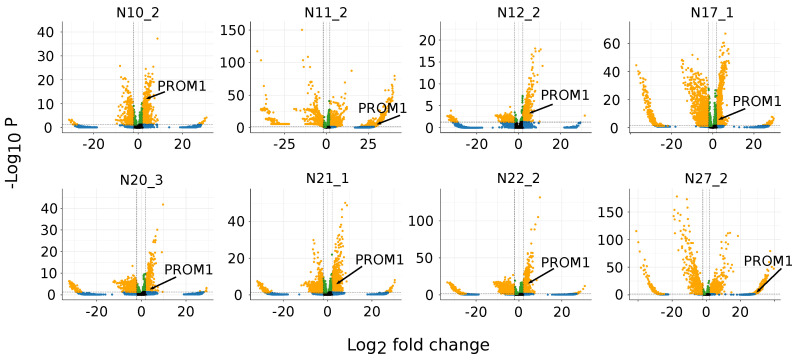
Volcano plot of the 8 clusters marked by *PROM1*. The number following the underscore of each title is the cluster ID. See Appendix A for the original names of the samples. The stippled lines represent the marker genes criteria: *p*-value < 0.05, and logFC > 2. Selected marker genes are the yellow dots on the right side of the volcano plot. For clusters N10_2, N11_2, N12_2, N17_1, N20_3, N21_1, N22_2, and N27_2, *PROM1* ranks 66/702 (top 9.4%), 5499/5901 (top 93.2%), 236/917 (top 25.7%), 1269/2181 (top 58.2%), 628/764 (top 82.2%), 315/891 (35.3%), 152/739 (top 20.6%), and 1524/2943 (top 51.8%), respectively.

**Figure 2 cancers-15-01557-f002:**
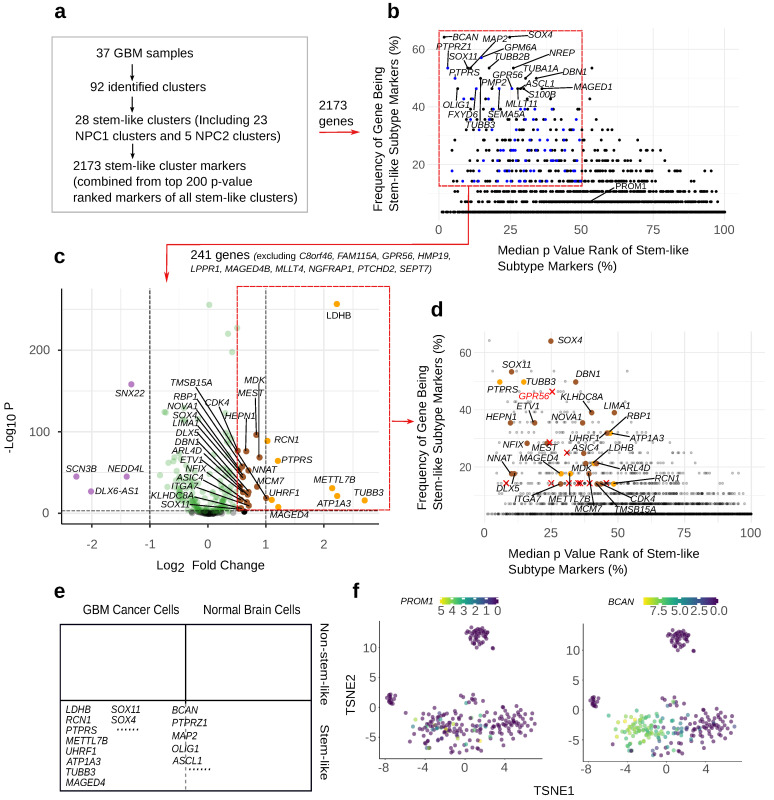
The procedure for the selection of stem-like markers of GBM that excel in both identifying stem-like subtypes within the tumor and being cancer-specific. (**a**) The process of selecting GBM stem-like subtype marker candidates. (**b**) The stem-like subtype marker candidates were evaluated based on their frequency of being a stem-like cluster marker and the corresponding significance shown by the median *p*-value across the represented clusters. The dots represent the 2173 unique stem-like marker candidates combined from the top 200 marker genes of the 28 stem-like clusters. (**c**) The expression level comparison between normal brain cells and GBM stem-like cells of 241 markers from the previous frequency–significance selection. The criteria for the orange genes are logFC > 1 and *p*-value < 0.01, and the criteria for the brown genes are 0.5 < logFC < 1 and *p*-value < 0.01. Both colors were considered significant in the comparison. (**d**) The genes selected to distinguish normal brain cells from GBM stem-like cells marked on the frequency–significance selection shown in Figure 2b. (**e**) Examples of the GBM stem-like markers that are cancer-specific and those that are not. *BCAN*, *PTPRZ1*, *MAP2*, *OLIG1*, *ASCL1*, etc. are non-cancer-specific. (**f**) Expression of *BCAN* and *PROM1* in the GBM stem-like subtype. This example is from sample N20.

**Figure 3 cancers-15-01557-f003:**
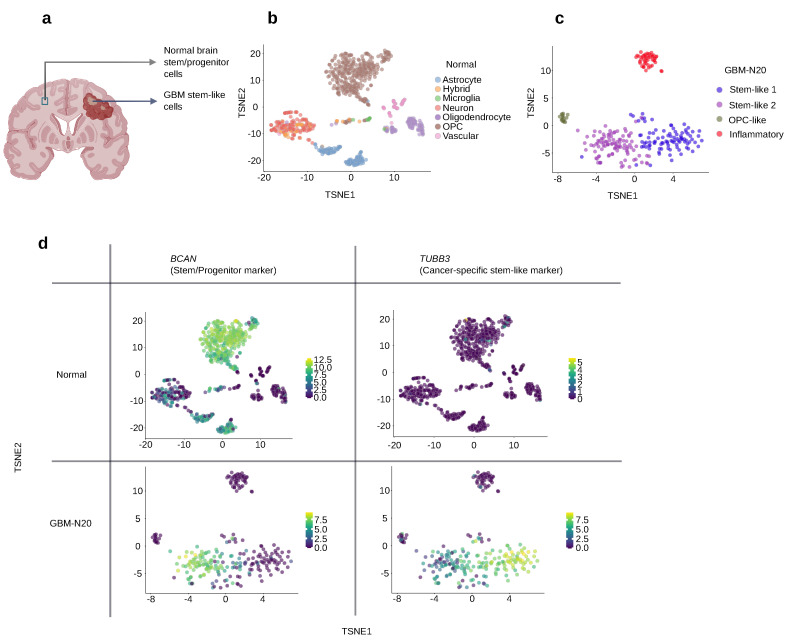
The expression of a general progenitor cell marker and cancer-specific stem-like marker for GBM. (**a**) Both GBM tumors and normal tissue possess progenitor cells. This illustration was created using BioRender.com. (**b**) TSNE for normal brain cells. (**c**) TSNE for the GBM cells from sample N20. (**d**) Comparing the expression of a general progenitor cell marker, *BCAN*, and a cancer-specific progenitor marker, *TUBB3*, between normal brain cells and GBM cells. The color bar with numbers represents the log-normalized counts of the gene.

**Figure 4 cancers-15-01557-f004:**
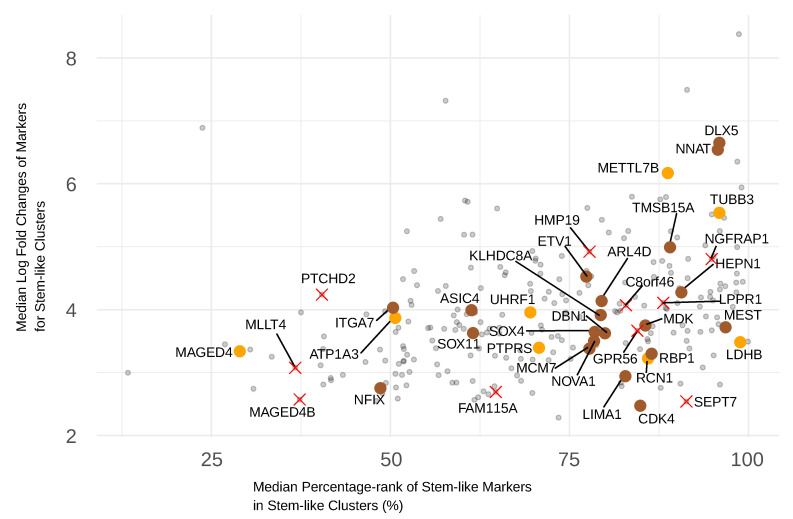
Selection of markers based on expression and overexpression in the stem-like subtype. The expression level of the gene is represented by the median percentage-rank, and the overexpression is shown by logFC obtained from the analysis of cluster markers. All the markers selected by comparison with normal brain cells are also marked using the same colors and shapes as in Figure 2c. The median was taken across all the cells in the stem-like clusters.

**Figure 5 cancers-15-01557-f005:**
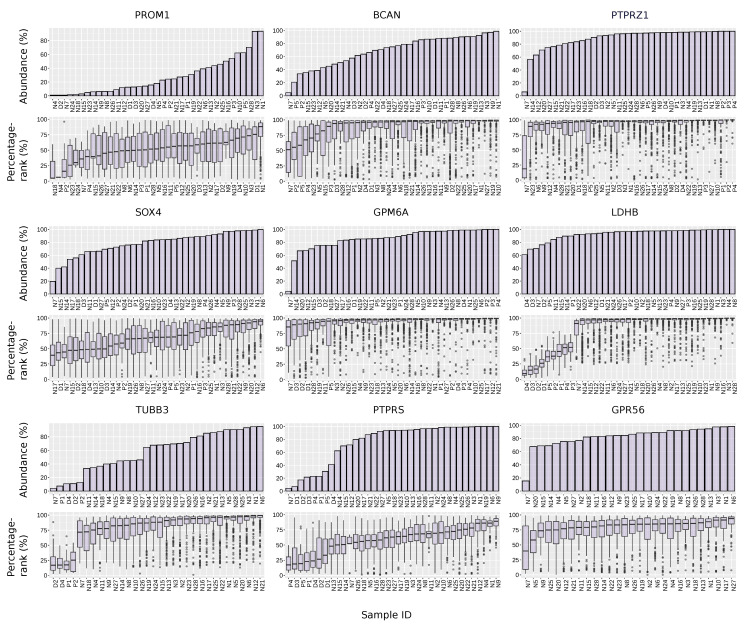
Abundance and percentage-rank of selected markers and *PROM1* across samples. The letters “N”, “P” and “D” in the sample ID represent samples from the studies of Neftel et al. [11], Patel et al. [24], and Darmanis et al. [25], respectively.

**Table 1 cancers-15-01557-t001:** Validation of stemness for highlighted markers.

Markers	Cell Subtype	Properties	Methods Used for Validation	References
*SOX4*	GSC	Stemness regulator, GSC signature marker, transcription factor (TF) highly expressed in embryonic, neural, or tumor stem cells	Transcriptome profiling, tumorigenesis in vivo	[37,38,48,49,50]
*SOX11*	GSC	GSC signature marker, stemness regulator	Transcriptome profiling	[37,51]
*ASCL1*	GSC	GSC signature marker	Transcriptome profiling, tumorigenesis in vivo and in vitro, genetic knock-down	[37,38,52,53]
*PTPRZ1*	GSC	Tumor initiating GSC marker, invasive GSC marker, overexpressed in stem-like phenotype of GBM spheroid	Tumorigenesis in vivo, genetic knock-down, invasion assays, tumorigenesis in vitro, transcriptome profiling	[54,55,56]
*BCAN*	pGSC	pGSC (proneural) signature marker, overexpressed in stem-like phenotype of GBM spheroid	Tumorigenesis in vitro, transcriptome profiling	[52,55,57]
*OLIG1*	GSC	Stemness regulator, GSC signature marker	Transcriptome profiling, tumorigenesis in vitro	[38,58,59]
*GPR56*	GSC	overexpressed in stem-like phenotype of GBM spheroid, neural stem cell marker, cancer stem cell (CSC) marker	Tumorigenesis in vitro, transcriptome profiling	[55,60,61]
*MAP2*	GSC	overexpressed in stem-like phenotype of GBM spheroid	Tumorigenesis in vitro, transcriptome profiling	[55]
*GPM6A*	GSC	GSC signature marker, invasive GSC marker	Tumorigenesis in vitro	[62]

## Data Availability

Publicly available datasets were analyzed in this study. The data can be found here: https://www.ncbi.nlm.nih.gov/geo/ (accessed on 1 January 2020) with accession numbers: GSE84465, GSE131928, GSE57872, GSE67835, and GSE84465.

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
