# Peer review of "Quantitative Evaluation of Stem-like Markers of Human Glioblastoma Using Single-Cell RNA Sequencing Datasets"

_cancers, 2023, doi:10.3390/cancers15051557_

Round 1

Reviewer 1 Report

The manuscript “Quantitative Evaluation of Stem-like Markers of Human Glioblastoma Using Single-cell RNA Sequencing Datasets” by Yue He et all is well-written and interesting.

Without doubts the study is descriptive, but the data provided are innovative, important and clinically feasible.

Major comments:

I would encourage the authors to validate their results experimentally in patient samples. Alternatively, they could correlate their results with clinical outcome of patients, using available online databases with clinical data.

Reviewer 2 Report

In their work, He and colleagues evaluated several GBM stem-like marker candidates by using existing single-cell RNA sequencing datasets, with the ultimate goal to identify the most relevant and robust biomarkers for GSC detection. In total, the research idea is quite interesting, the research protocol is well-designed and the manuscript is well-written and comprehensible. I would only like to propose the following.

1. The authors compared GSCs with normal brain cells and suggested a panel of 8 cancer-associated proteins (LDHB, RCN1, PTPRS, METTL7B, UHRF1, MAGED4, ATP1A3, and TUBB3) for therapeutic targeting. It would be interesting to present TCGA data regarding patient survival based on the expression of these genes, both separately and as a signature. These data, if relevant, would strengthen the value of the proposed markers.

2. The authors should further discuss the relevant roles of the proposed genes in the discussion section.

Reviewer 3 Report

In this paper, the authors identified some marker genes of human glioblastoma by integrating several datasets. The following lists some concerns. First, for the marker genes, it will be much better to use them for classification or other special task of these markers. Second, the authors integrated several datasets from different studies. The batch effects need be removed for the integration. It is an alternative to use one data for identifying these marker genes, and validate them in the other data. The identified marker genes have no validations. The claim of these markers need be verified and the enriched functions need be presented. The authors claimed these important analyses. However, the results contain few information. And last but not least, the single cells are diverse. It is not clear about the cell types of these marker genes. The cell-specified markers seem be important and the information need be contained.

Reviewer 4 Report

Interesting and very well written study regarding the Quantitative Evaluation of Stem-like Markers of Human Glioblastoma Using Single-cell RNA Sequencing Datasets. 

I would just suggest to the authors to better rewrite conclusions, maybe in a simple manner, in order to be more understandable even to researchers not familiar with RNA sequencing datasets. 

Round 2

Reviewer 1 Report

The authors have adequately addressed the comments in the revised version of the manuscript. Therefore, I have no further comments.